# Preparation and Characterization of Glass-Ceramic Foam from Clay-Rich Waste Diatomaceous Earth

**DOI:** 10.3390/ma15041384

**Published:** 2022-02-13

**Authors:** Martin Sedlačík, Martin Nguyen, Tomáš Opravil, Radomír Sokolář

**Affiliations:** 1Materials Research Centre, Faculty of Chemistry, Brno University of Technology, Purkyňova 464/118, Brno 61200, Czech Republic; opravil@fch.vut.cz; 2Faculty of Civil Engineering, Institute of Technology of Building Materials and Components, Brno University of Technology, Veveri 331/95, Brno 60200, Czech Republic; nguyen.m@fce.vutbr.cz (M.N.); sokolar.r@fce.vutbr.cz (R.S.)

**Keywords:** foam glass-ceramics, nepheline, porous materials, secondary materials, waste diatomaceous earth

## Abstract

In this study, the potential use of waste diatomaceous earth from the production of diatomaceous earth for filtration purposes, as an alternative raw material for foam glass production, was explored. The chemical and mineralogical composition and the high temperature behavior of waste diatomite were studied to assess its suitability for foam glass production. Glass-ceramic foams were prepared using NaOH solution as a foaming agent, via a hydrate mechanism. The influence of different pretreatments and firing temperatures on the foam’s structure, bulk density and compressive strength was investigated. High temperature behavior was studied using TG/DTA analysis and high temperature microscopy. Phase composition was studied using X-ray diffraction analysis. Glass-ceramic foam samples of a high porosity comparable to conventional foam glass products were fabricated. The pretreatment temperature, foaming temperature and sintering holding time were found to have a significant influence on foam properties. With increased pretreatment temperature, pyrogenic carbon from the thermal decomposition of organic matter contained in the raw material acted as an additional foaming agent and remained partially unoxidized in prepared foams. The bulk densities of prepared samples ranged from 150 kg/m^3^ to 510 kg/m^3^ and their compressive strengths were between 140 and 1270 kPa.

## 1. Introduction

Foam glass is an inorganic thermal insulation material which was first introduced in the 1920s. Today, mass production of foam glass for thermal insulation purposes is conducted exclusively using a powder sintering method due to its lower energy consumption and higher production capacity compared to other methods of porous glass preparation, such as direct melt foaming, sol-gel processes, high-pressure physical foaming or leaching of phase separated glasses. In this process, waste glass cullet or specially prepared and finely milled glass powder and a suitable foaming agent, are mixed and heated in a furnace [1]. Increasing temperature causes the sintering of glass particles, the subsequent formation of a glass melt and the release of gaseous reaction products due to foaming agent reaction or thermal decomposition [2]. The porous structure is then fixed via rapid cooling and the resulting foams are either annealed and slowly cooled to allow for dissipation of stress in the case of foam glass blocks, or rapidly cooled to room temperature to cause the cracking and disintegration of the material to produce lightweight foam glass aggregate [1,2,3,4].

Increasing requirements on insulation materials in terms of their properties and environmental impact, and concerns regarding the fire-resistance of polymer insulation, mean that foam glass is becoming increasingly popular due to its long service life, thermal, mechanical and chemical resistance [5]. In recent years, foam glass production has experienced a steady growth with more than 1.5 million tons produced worldwide, and production is expected to reach almost two million tons by 2024 [6].

Because of the relative scarcity of suitable glass cullet in some regions, the high price of foam glass, and efforts to lower the environmental impact of building materials and achieve zero waste production, a significant amount of work is ongoing to discover alternative raw materials for the production of foam glass [7].

In the past, most of the focus in this area has been on glass that is difficult to recycle, mainly cathode-ray tube glass [8,9,10], solid waste from incineration processes [11,12,13], red mud [14,15] and various types of slags [16,17]. In terms of natural raw materials, in addition to zeolitic rocks [18,19], diatomaceous earth has been recognized as a suitable candidate due to its price, availability and the stability of the mineralogical composition of its deposits. However, most of the available studies have focused on either a partial replacement of glass in ordinary foam glass production or the use of high quality raw diatomaceous earth with a high diatom shell content [7,20,21].

Commercially produced foam glass is currently almost exclusively produced using carbonaceous foaming agents like soot, silicon carbide, or glycerol [1]. However, several other foaming agents have either been used in the past or their use has been studied at a laboratory scale, including AlN, TiN, MnO_2_, Mn_3_O_4_, CaCO_3_ and other carbonates and sulfates [22,23,24,25]. Recently, increasing attention has focused on the use of NaOH or sodium silicates as foaming agents. NaOH is especially interesting in conjunction with alternative raw materials for foam glass production because it can react with amorphous silica and clay minerals to form silicate hydrates. The formation and subsequent thermal decomposition of these hydrates cause a significant decrease in the temperature for sintering, melt formation and foaming caused by the release of water vapor from structurally bound water [7,26,27].

In comparison, this study explores the potential of solid waste generated during centrifugal classification and filter pressing, during filter diatomaceous earth production. This waste material is composed primarily of residual water, clay particles and broken diatom shells with particle sizes less than 10 µm. This byproduct of filter diatomaceous earth production does not yet have an industrial application and is used primarily for backfilling surface mines. The manufacturing of insulation materials using exclusively diatomaceous earth production waste, without the need for glass cullet use, would significantly contribute to lowering the environmental impact of this industry and represents another step in the zero-waste production process. Achieving independence on the glass recycling industry is also desirable for the potential streamlining of the production process and lowering production and transportation costs and potential complications related to supply chain issues. This approach also avoids common complications related to foam glass production from waste glass cullet, such as reliance on developed glass recycling chains and the fluctuating chemical composition of glass cullet.

The aim of the present investigation is the thorough characterization of the chemical and mineralogical composition and thermal behavior of this interesting resource in the context of foam glass production and the preparation and characterization of sintered glass-ceramic foams via a hydrate mechanism using sodium hydroxide as a foaming agent.

## 2. Materials and Methods

The raw material used in this study was the solid waste byproduct from filter diatomaceous earth production from the Borovany deposit. It is a material with a particle size under 10 µm by centrifugal classification, and is mostly composed of clay mineral impurities (identified as kaolinite Al_2_O_3_·2SiO_2_·2H_2_O) and broken diatomite shells.

### 2.1. Sample Preparation

To allow for thorough mixing with foaming additives, waste diatomite was processed into dry powder. Raw material in the form of filter press plates was first broken up into smaller pieces and dried at 105 °C to a constant mass. Then it was crushed using a laboratory jaw crusher and milled for 15 s using a vibration mill to break up any residual agglomerates of fine particles.

Dry milled powder was mixed with 50 wt.% NaOH solution in the ratio NaOH: solid of 30:100. The mixture was homogenized by manual stirring, divided into three separate closed polypropylene containers, and placed into an electrical oven preheated to 50 or 80 °C or left to react at laboratory temperature. The mixture was allowed to react for 1h at the set conditions to intensify the reaction between the constituents. After 1 h the green body pellets were pressed using a laboratory hydraulic press with a steel mold. Pellets were 4 cm in diameter, the pressure was 5 MPa and the pressing time was 30 s. Pressed pellets were then fired at temperatures of 800, 850 or 900 °C for 30 min at atmospheric conditions in a canthal furnace with a heating rate of 5 °C/min and cooling rate of 5 °C/min down to 600 °C. Samples were then left to cool to laboratory temperature. A total of three samples were made for each combination of preparation and firing conditions. The firing curves used are shown in Figure 1. The influence of different sintering holding times on the resulting properties was also evaluated. The sintering holding times were chosen as 15, 30 and 45 min at 850 °C with the same heating and cooling rate as described above.

### 2.2. Characterization Methods

The density of raw waste diatomite was determined using a Hubbard pycnometer, 1 g of dried powder sample and isopropanol. The stated density value is an average of three separate measurements. Loss on drying (LOD) and loss on ignition (LOI) were determined according to ČSN 72 0102 and ČSN 72 0103 standards, respectively.

The oxide composition of waste diatomaceous earth was analyzed using a Vanta energy dispersive X-ray fluorescence (XRF) spectrometer. Elements with proton numbers ranging from those of magnesium to uranium were analyzed and lighter elements were defined as light elements (LE). Phase composition was analyzed using X-ray diffraction (XRD) and carried out using a Bruker D8 Advance diffractometer with a Cu anode (λK_α_ = 1.54184 Å) and an input current of 30 mA and variable divergence apertures at θ–θ reflective Bragg–Brentano parafocusing geometry with the diffraction angle 2θ in the range of 4.5° to 90° with a step scan of 0.01°. Determination of the amorphous phase content was carried out using 20 wt.% CaF_2_ addition as a standard.

Particle size distribution was determined using laser diffractometry (Helos KR, Sympatec, Clausthal-Zellerfeld, Germany). While the particle size of milled powder was determined directly in the dry state, the particle size of the raw material was determined in a water suspension. The suspension was prepared using 2 g of waste diatomite and 50 cm^3^ of water. The suspension was placed into an ultrasonic bath to ensure full deagglomeration of particles. Specific surface area was determined using a BET analyzer (Quantachrome Nova, Boynton Beach, FL, USA).

The high temperature behavior of the material was studied using thermal gravimetry and differential thermal analysis (TG-DTA) (SDT 650, TA Instruments, New Castle, DE, USA). TG-DTA measurement was carried out using a 25 mg sample with a heating rate of 10 °C/min up to 1400 °C in air atmosphere and a fused alumina crucible. Sintering and foaming behavior was monitored via heating microscopy (EM 201, Hesse Instruments, Osterode am Harz, Germany) with a heating rate of 5 °C/min up to 1500 °C or reaching the flow temperature of the measured sample.

For the determination of the bulk density of the prepared foam, samples were cut and filed into regular shapes, the dimensions were measured, and the bulk density (kg/m^3^) was calculated using their mass and volume, using the following formula:ρ = m/(a·b·c)(1)

ρ–bulk density [kg/m^3^]

m–sample weight [g]

a, b, c–measured sample dimensions [mm]

Compressive strength was measured on cubic samples using the tension and compression testing machine, Instron 5985 (Instron, Norwood, MA, USA), 250 kN with a displacement rate of 1 mm/min. The compressive strength of each sample was determined as the maximum load of the first plateau of the stress-strain curve divided by the cross-sectional area of the measured sample.

Photographs of cross sections of prepared foam for pore structure morphology evaluation were acquired using a Nikon D5100 digital camera (Nikon, Minato, Japan).

Bulk density and compressive strength were measured for three samples and the average values for each sample are stated.

## 3. Results and Discussion

### 3.1. Raw Material Characterization

The oxide composition of waste diatomite determined by the XRF analysis is shown in Table 1.

LOD, LOI, and density of waste diatomite are listed in Table 2. The raw material had values of D_50_ = 4.56 µm and D_90_ = 11.11 µm. Dried crushed and milled waste diatomite powder used for foam glass production had D_50_ and D_90_ values of 2.39 and 6.39 µm, respectively. The specific surface area of used diatomite powder was determined to be 49191 m^2^/kg, using the BET method.

The results of the XRD analysis of the raw material are shown in Figure 2. The XRD pattern shows a significantly elevated background, indicating the high amount of non-crystalline phases present in the sample. Major diffraction peaks correspond to kaolinite, low quartz and anatase.

The results of the XRD measurement with CaF_2_ addition used to quantify the amorphous phase content are shown in Table 3. The major phases in waste diatomaceous earth were identified as a kaolinite phase with the basal spacing d_001_ of 0.7134 nm and an amorphous phase. The remaining components constitute less than one percent of the material and were identified as low quartz, anatase and calcite. A high amorphous phase content is desirable for foam glass production and is indicative of diatomite presence in waste material. However, presence of kaolinite in such large quantities means that simple melting of the sample using a conventional reduction or neutralization foaming agent and converting it into foam glass is not a viable option.

The results of the thermal analysis of the sample are shown in Figure 3. During heating, the sample showed a total weight loss of 10.53%.

The first major weight loss occurred in the temperature range up to 200 °C and was due to the release of physically bound water in raw diatomite [28]. The second weight loss in the temperature range 250–700 °C, with a distinct visible peak in the heat flow curve, resulted from dehydroxylation of kaolinite according to Equation (2) [29]. Part of the weight loss in this temperature range can also be attributed to the burn out of organic matter present in the raw material [28].
Al_2_O_3_·2SiO_2_·2H_2_O → Al_2_O_3_·2SiO_2_ + 2H_2_O(2)

The weight loss at 619 °C was likely due to the loss of structural water by the dehydroxylation of silanol groups in the opal structure of diatomite [7,28].

The dehydroxylation process continued up to 1400 °C, resulting in an additional weight loss of 0.78% in the temperature range 800–1400 °C [28]. The three exothermic peaks which are visible at 967 °C, 1142 °C and 1187 °C are likely caused by the crystallization of new phases.

### 3.2. Foam Preparation and Characterization Results

Results of the high temperature microscopy measurement of raw diatomite and diatomite mixed with NaOH solution are depicted in Figure 4 as an A/A_0_ area change of the sample silhouette. Comparison of sample shape change upon heating can be seen in Figure 5.

Raw waste diatomite underwent a sintering process starting at around 800 °C accompanied by continuous shrinkage of the sample. A plateau can be seen at the heating microscopy area curve in the temperature range 1150–1300 °C, caused by the crystallization of the sample. This claim is supported by correlation with the exothermic peaks visible in the TG-DTA measurement. Shrinking due to sintering continued up to 1400 °C. However, significant deformation due to the melting of the sample was not observed.

In comparison, reaction with NaOH solution significantly altered the high temperature behavior of used waste diatomite. A small expansion can be seen in the temperature range 300–700 °C. This is presumably caused by the release of physically and weakly bound water contained in the reaction products. The decrease in sample area around 700 °C can be attributed to the sintering and melting of the sample. The foaming process starts at 750 °C as a result of decreasing melt viscosity with increasing temperature. Maximum expansion is achieved at 850 °C followed by a short plateau and then a sharp decrease in sample area indicating rupture of pore walls leading to coalescence, gas release and the collapse of the foam. The reaction of waste diatomaceous earth with NaOH causes a decrease in the onset temperature of the sintering process by 100 °C and allows the melting of the sample necessary for a successful high temperature foaming process.

#### 3.2.1. Structural Study

Figure 6 represents the mineralogical composition of the fired foam glass samples. The only crystalline phase present in all samples which sintered at various temperatures is nepheline (Na_3_KAl_4_Si_4_O_16_). Considering kaolinite and amorphous diatomite were two major constituents of waste diatomaceous earth, crystallization of nepheline is a result of the decomposition of aluminosilicate hydrates. Cristobalite, commonly found in the vast majority of glass-ceramic foams, was not present in foamed waste diatomite samples. Minor constituents present in the raw material were not apparent in the XRD pattern of foamed samples and thus were likely incorporated into the glassy network of the amorphous phase. The background curvature at low 2θ angles indicates the presence of an amorphous glass phase and the background noise (scattering) indicates the presence of iron compounds due to the interference with the Cu Kα radiation source [30].

Figure 7a–f. represents the SEM microphotographs of foamed waste diatomite samples. The overall pore structure is presented in Figure 7a–c with a 100× magnification for all firing temperatures. The microstructure inside the pore structure is presented in Figure 7d–f with a 2000× magnification for all firing temperatures. It is apparent from Figure 7a–c that with the increasing firing temperature, the pore size is greater. Figure 7d–f displays small nepheline crystals which crystallized on the inside wall of the pores.

Images of the foamed waste diatomite samples with their respective scale can be seen in Figure 8. At all pretreatment reaction temperatures, an increase in pore size with increasing foaming temperature can be observed. This is the result of two phenomena. Firstly, viscosity of the melt decreases with the increase in temperature, and, secondly, the pressure of water vapor released from the thermal decomposition of formed aluminate silicate hydrates increases. Together, these phenomena cause the expansion of the melt and the emergence of a porous structure. If the pressure of gas contained inside the pores becomes too high, the pore walls become thinner and eventually rupture, causing the coalescence of the two pores and the densification of the pore walls.

Color change of the sample core can be seen with increasing pretreatment temperature. This change is a result of the thermal decomposition of organic matter present in the raw material. With increasing reaction temperature, the rate of reaction between alkali and both kaolinite and amorphous opal SiO_2_ from broken diatom shells increases, and a higher percentage of the material is converted into sodium aluminate silicate hydrates. In samples reacting at laboratory temperature before firing, most of the organic matter decomposes before the closed porosity is reached, while with increasing pretreatment temperature, more pyrogenic carbon is created due to the reduced availability of oxygen during the thermal decomposition of the organic matter. The amount of oxidizing agents present is then insufficient for the full oxidation of formed carbon, causing the black coloration of the foam core. This opens possibilities for further foaming and decreasing the bulk density using additional oxidizing agents in the raw mixture to take advantage of the oxidizable carbon present in waste diatomite foams.

Heterogenous pore size distribution was observed in all samples, mainly caused by the significant coalescence of pores observable in samples fired at higher temperatures. Considering the presence of multiple reactive species in the raw material, this could also suggest inhomogeneous pore evolution due to the uneven reaction of hydroxide with waste diatomite. Compared to ordinary foam glass production, where there is typically only one foaming agent present in homogenous glass cullet, multiple present phases with different reactivities are likely contribute to a heterogeneous pore structure.

#### 3.2.2. Bulk Density and Compressive Strength

The bulk densities of the fired samples can be seen in Figure 9. The bulk density of samples foamed at 800 °C was highest out of all of the foaming temperatures and was in the range 380–510 kgm^−3^.

The lowest bulk density for all reaction temperatures was achieved using an 850 °C foaming temperature. The lowest bulk density out of all of the prepared samples was achieved using an 80 °C pretreatment and an 850 °C foaming temperature; this achieved a bulk density of 150 kgm^−3^. A further increase in foaming temperature led to an increase in the bulk density of all samples. Coupled with a highly inhomogeneous pore size distribution this can all be attributed to the high coalescence of particles, the subsequent densification of pore walls and the partial escape of foaming gas from the melt. This is in good agreement with heating microscopy measurements using small sized samples. However, the expansion of larger samples was significantly higher, presumably due to their higher volume to surface area ratios which allow less of the foaming gas to escape from the melt. An elevated reaction temperature during sample preparation generally caused a decrease in the bulk density of prepared foams foamed at identical temperature and holding times.

Figure 10 shows the compressive strength of all fired samples. Generally, the compressive strength of samples correlates with their bulk density.

With decreasing bulk density, the compressive strength of the sample also decreases. However, samples which reacted at 50 °C and 80 °C and were fired at 900 °C reached lower values of compressive strength despite their higher density, compared to samples fired at 850 °C. This was caused by the extensive coalescence of pores, causing an increase in pore size and irregularities in both their shape and size distribution. The highest compressive strength was reached in samples fired at 800 °C, which corresponded with the highest bulk density and also the lowest average pore size. The compressive strength of prepared foams ranged between 140 and 1270 kPa. The lowest bulk density values obtained using waste diatomite utilized in this study exceed similar studies utilizing high quality raw diatomite [7,21,31] and are comparable to those of commercially produced foam glass. However, further research is required in order to improve pore structure homogeneity and mechanical properties.

#### 3.2.3. Influence of Sintering Holding Time on the Properties

The influence of different sintering holding times on the bulk density and compressive strength of the test samples was examined. Sintering holding times were experimentally chosen as 15, 30 and 45 min with a maximum sintering temperature of 850 °C.

As can be seen in Figure 11, the bulk density is lowest at the sintering holding time of 30 min in all different pretreatment cases of the samples. Low bulk density leads to better thermal insulation properties which are important attributes for the glass-ceramic foams. Overall, the difference between the pre-treated samples at 50 °C and 80 °C fired at different sintering holding times was minimal.

Figure 12 represents the influence of different sintering holding times on the compressive strength. As can be seen, the highest compressive strength was achieved by firing at 850 °C with a sintering holding time of 15 min, however the standard deviations of these foam samples were very high. This was due to the short sintering holding time which was insufficient for the foam structure to form and develop. The values of compressive strength and bulk density are in correlation with the pore structure. The optimal sintering holding time for the optimal resulting properties of glass-ceramic foam for use as a thermal insulation is 30 min.

## 4. Conclusions

Waste diatomite from filter diatomaceous earth production was thoroughly characterized with respect to its high temperature behavior and suitability for lightweight foam glass-ceramic production. Waste diatomite underwent drying, crushing, and milling processes to obtain a dry powder precursor. Glass-ceramic foam samples were then fabricated via a sintering method using NaOH as a foaming agent. Due to the nature of studies dealing with production waste utilization, further research and the modification of the proposed methods would be required for different deposits. However, a road map for the utilization of waste diatomite with a very high clay content, was developed. Apart from altering the high temperature of the material and allowing foam glass production, using NaOH as a foaming agent altered the behavior of waste diatomite in the green state from powder to a moldable plastic state due to partial conversion to hydrated phases. Compared to ordinary foam glass production process, among other things, this opens granulation possibilities and increases the manipulation strength of green bodies without the requirement of excessively high pressures. In addition, no toxic products i.e., H_2_S, CO are released during foaming process.

The influence of pretreatment and foaming temperatures on foam structure, bulk density and compressive strength was investigated. Reaction with NaOH lowered the sintering and melting temperature, which enabled the foaming of waste diatomite at a temperature range similar to that of conventional foam glass. Only crystalline phase present in foamed diatomite samples was found to be nepheline (Na_3_KAl_4_Si_4_O_16_). Increased pretreatment temperature caused the foam sample cores to change color to black. This color change was caused by unoxidized carbon resulting from the thermal decomposition of organic matter present in the raw waste diatomite, which has not previously been mentioned in similar studies. This opens further possibilities for further porosity increase using additional oxidating agents. The bulk density of prepared foams ranged from 150 kg/m^3^ to 510 kg/m^3^, and their compressive strengths ranged from 140 kPa to 1270 kPa. The foams with the highest bulk densities and compressive strengths were obtained using a foaming temperature of 800 °C. Samples foamed at 800 °C and pretreated at laboratory temperature achieved a compressive strength of 1.27 MPa with a bulk density of 510 kg/m^3^. The lowest bulk density (150 kg/m^3^) was achieved using an 80 °C pretreatment temperature and an 850 °C foaming temperature, with a corresponding compressive strength of 160 kPa. Waste diatomite proved to be a suitable alternative raw material for foam glass-ceramic production via the hydrate mechanism. This provides an alternative approach for foam glass production independent of the glass recycling industry in areas with a high amount of diatomite production. It also provides an outlet for turning a currently unutilized secondary product into a thermal insulation material.

## Figures and Tables

**Figure 1 materials-15-01384-f001:**
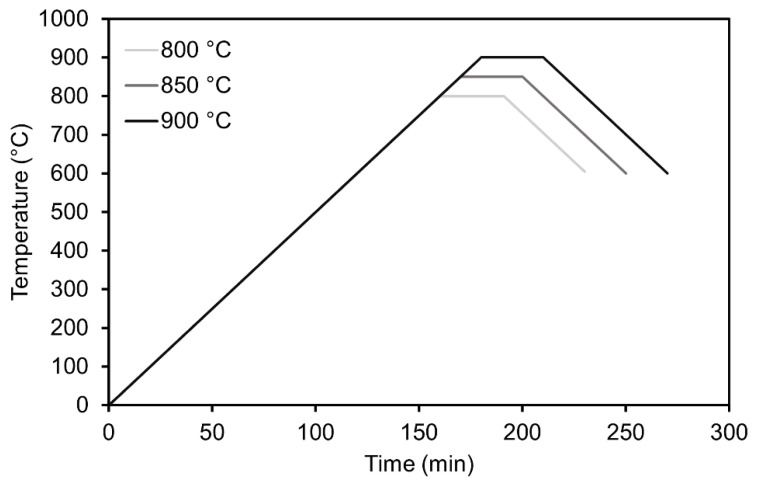
Firing curves used for samples foamed at 800, 850 and 900 °C.

**Figure 2 materials-15-01384-f002:**
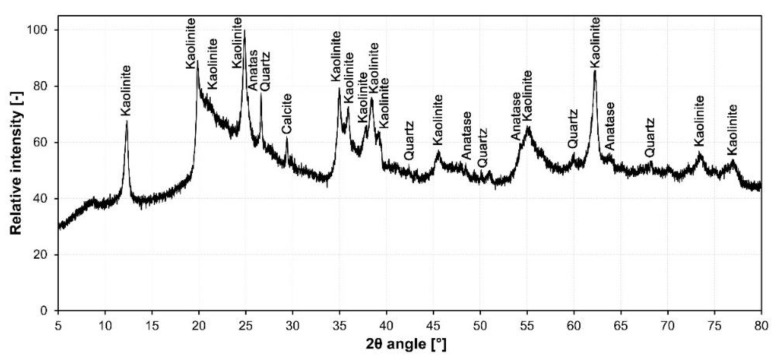
XRD pattern of waste diatomaceous earth.

**Figure 3 materials-15-01384-f003:**
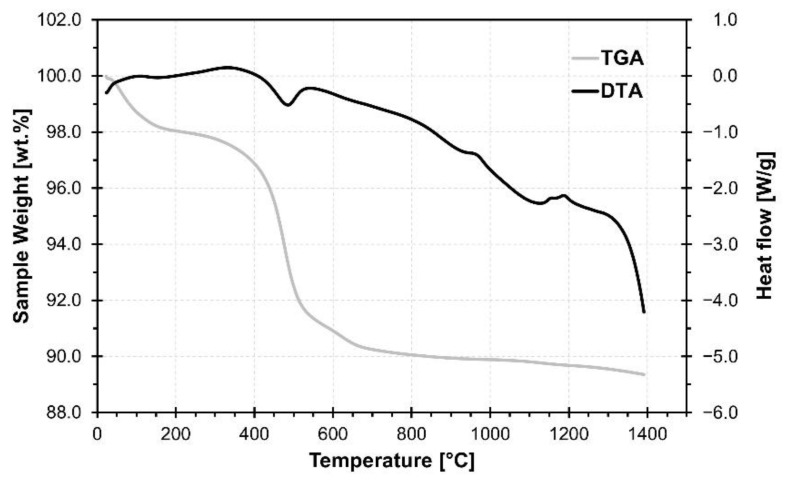
Results of TG/DTA analysis of waste diatomaceous earth.

**Figure 4 materials-15-01384-f004:**
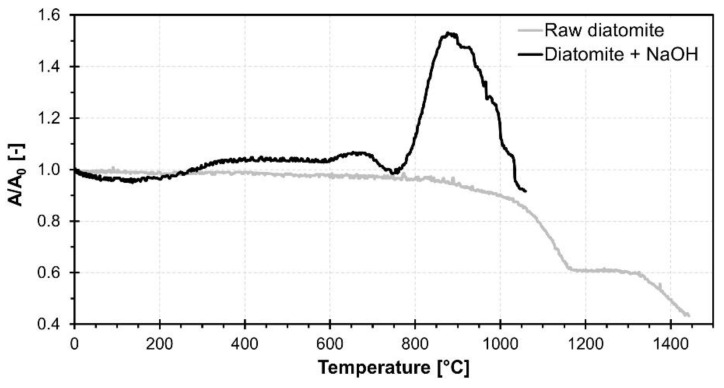
Results of heating microscopy measurement of dried raw diatomite powder and diatomite mixed with NaOH solution.

**Figure 5 materials-15-01384-f005:**
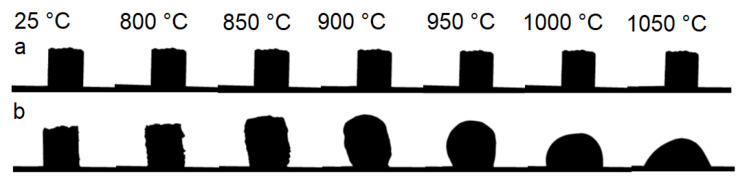
Comparison of sample shape change upon heating. (**a**) raw diatomite, (**b**) diatomite +NaOH.

**Figure 6 materials-15-01384-f006:**
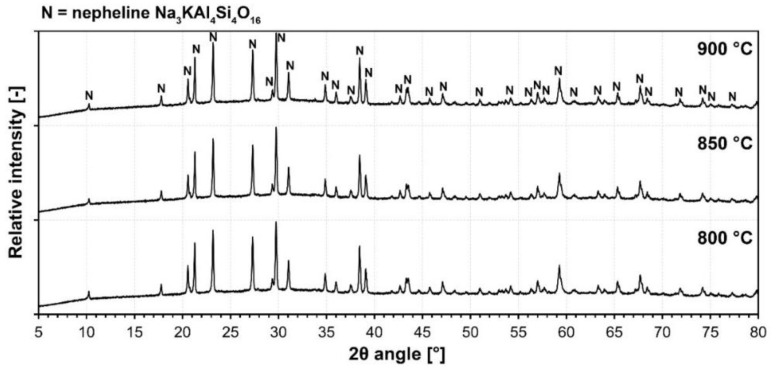
Examples of XRD patterns of waste diatomite foams fired at different temperatures (samples with pretreatment temperature 50 °C shown).

**Figure 7 materials-15-01384-f007:**
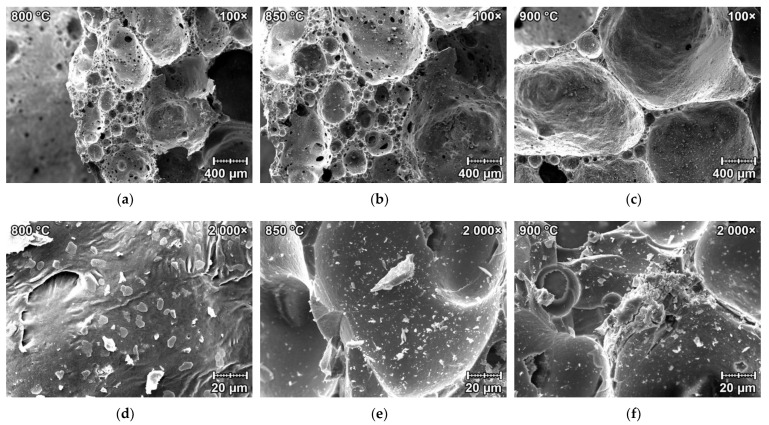
SEM microphotographs of the foamed waste diatomite samples: (**a**) sample fired at 800 °C with 100× magnification; (**b**) sample fired at 850 °C with 100× magnification; (**c**) sample fired at 900 °C with 100× magnification; (**d**) sample fired at 800 °C with 2000× magnification; (**e**) sample fired at 850 °C with 2000× magnification; (**f**) sample fired at 900 °C with 2000× magnification.

**Figure 8 materials-15-01384-f008:**
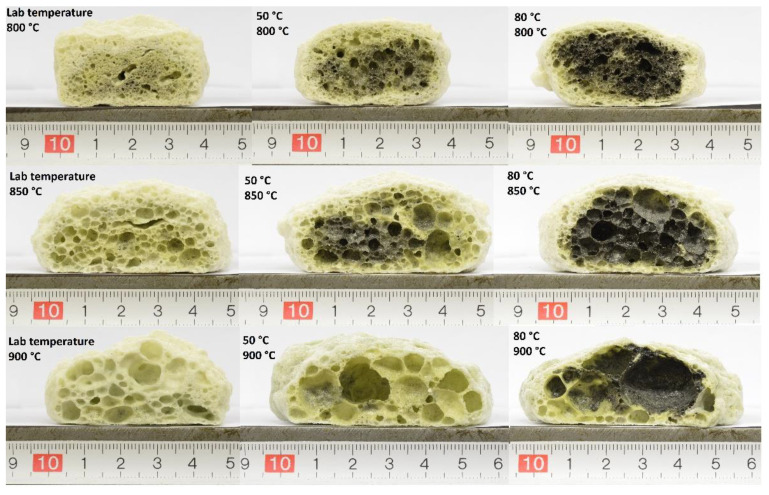
Porous structure of prepared glass-ceramic foams with their respective scale (in cm).

**Figure 9 materials-15-01384-f009:**
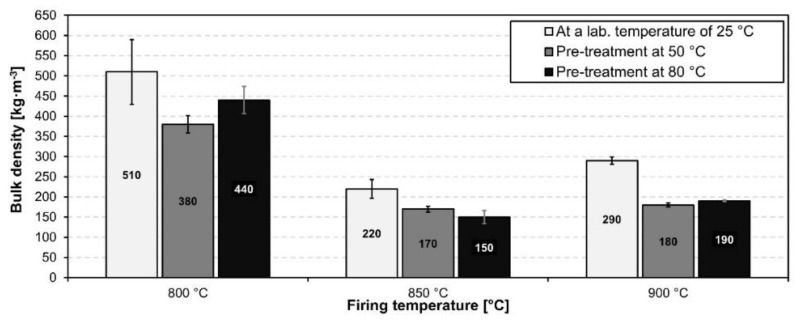
Bulk density of prepared foam samples fired at different temperatures.

**Figure 10 materials-15-01384-f010:**
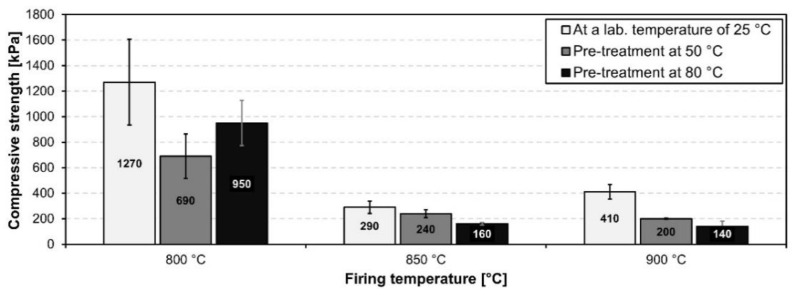
Compressive strength of prepared foam samples fired at different temperatures.

**Figure 11 materials-15-01384-f011:**
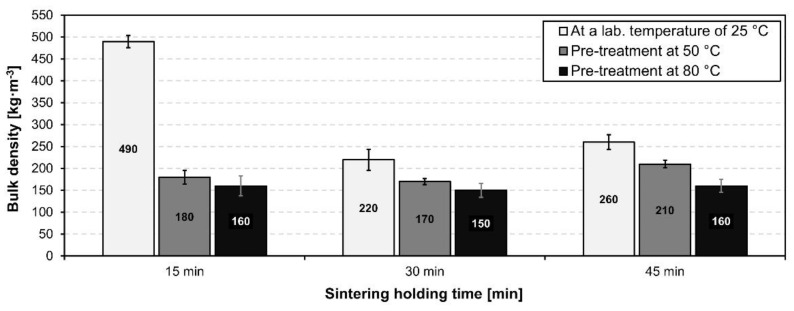
Bulk density of prepared foam samples fired at 850 °C with different sintering holding times of 15, 30 and 45 min.

**Figure 12 materials-15-01384-f012:**
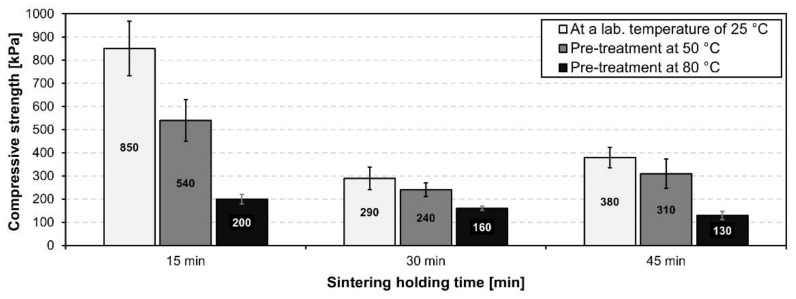
Compressive strength of prepared foam samples fired at 850 °C with different sintering holding times of 15, 30 and 45 min.

**Table 1 materials-15-01384-t001:** Oxide composition of waste diatomaceous earth.

Oxide	SiO_2_	Al_2_O_3_	Fe_2_O_3_	TiO_2_	CaO	K_2_O	CuO	LE
Amount (wt.%)	63.9	18.1	3.3	0.8	0.7	0.3	0.2	12.7
SD (wt.%)	0.1	0.1	0.01	0.01	0.01	0.01	0.01	0.1

**Table 2 materials-15-01384-t002:** LOD, LOI and density of waste diatomaceous earth.

LOD (wt.%)	LOI (wt.%)	Density (g·cm^−3^)
45.1	10.7	0.7988

**Table 3 materials-15-01384-t003:** Phase composition of waste diatomite.

Phase	wt.%
Kaolinite	59.9
Quartz low	0.2
Anatase	0.5
Calcite	0.2
Amorphous	39.2

## Data Availability

The data presented in this study are available on request from the corresponding author.

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
