# Peer review of "Preparation and Characterization of Glass-Ceramic Foam from Clay-Rich Waste Diatomaceous Earth"

_materials, 2022, doi:10.3390/ma15041384_

Round 1

Reviewer 1 Report

he authors presented the results of studies on production  of Glass-Ceramic Foam from Clay-Rich Waste Diatomaceous Earth.

The aim was to use of waste diatomaceous earth from production of diatomaceous earth for filtration purposes, as an alternative raw material for foam glass production, was explored.

The topic of the article is up to date, the introduction and literature survey is easy detailed, the authors discussed already all available literature sources. This problem is relevant for journal scope. 
The concept and aim are clearly defined.  The presentation and discussion of the result is clear and very detailed. The conclusions are well extracted from the results and discussion. The manuscript follows the formal regulations of MDPI journals.
I suggest the acceptance after minor revision.
Other weaknesses to be corrected:
Keywords should be in alphabetical order.
Explain the formula (1) - applies to the names a, b, c
Please improve the manuscript with a English proofreading

Reviewer 2 Report

The manuscript ,,Preparation and Characterization of Glass-Ceramic Foam from Clay-Rich Waste Diatomaceous Earth,, has interesting findings. However, the manuscript needs to be improved before publication can be considered.

Please be sure that your manuscript thoroughly establishes how this work is fundamentally novel. Specific comparisons should be made to previously published materials that have a similar purpose. Please present a strong case for how this work is a major advance. This needs to be done in the manuscript itself, not just in the response to review comments. 

Please be sure that your abstract and your Conclusions section not only summarize the key findings of your work but also explain the specific ways in which this work fundamentally advances the field relative to prior literature.

The abstract  should be factual. It should contain objectives, results and conclusions.

The significance of this study should be more emphasize in the introduction.

A look at these papers may be helpful. https://www.sciencedirect.com/science/article/pii/S0167577X14021259

 https://www.sciencedirect.com/science/article/pii/S0169131719301413

Line 81: Give the structural formula of the clay from this deposit. 

Line 105: Why did you use isopropanol as a solvent?

Line 112: Indicate the range of 2 theta angles, in which were XRD measurements were performed.

Line 141: What were the standard deviations?

Line 152: And what about the value of d001 diffraction, the so-called basal spacing, which talks about the size of the interlayer space. Size is given in nm. This is also important information that XRD can provide us. 

Line 162: It can be seen from the table that there is a large amount of amorphous phase in the sample. This is difficult to distinguish with XRD. Why didn't you perform an FTIR analysis that could identify individual functional groups.

Line 273:  Indicate the possible risks of such research. Add your recommendations for future research.

Line 305: Make sure the references are added correctly according to the journal's instructions.

Reviewer 3 Report

This paper generally provides insight for the development of “ Preparation and Characterization of Glass-Ceramic Foam from Clay-Rich Waste Diatomaceous Earth”. There is a nice approach followed and analyzed. However, it lacks explanation and comparison that need to justify the as prepared samples.

  1. The properties of the as obtained foam glass-ceramics should be compared with those of the foam glass-ceramics in other reports.
  2. Some writing errors exist, such as “TiN MnO2” in line 60, “The Mixture” in line 90, “con-crystalline” in line 149, “hydrates In” in line 233, etc.
  3. SEM images of the samples should be provided.
  4. The non-uniform pore size of the samples should be explained.
  5. The effects of sintering holding time on the properties of the samples should be provided.

Reviewer 4 Report

Line 30: „it (foam glass) is exclusively produced using powder sintering method” This sentence should be clarified as other methods (e.g. sol-gel direct foaming, rapid prototyping) are also used for the production of foam glasses (see more: https://doi.org/10.1533/9780857097033.2.191 ).

Line 89: „Dry milled powder was mixed with 50% NaOH solution – It should be clarified, what type of % exactly”

Figure 3: labels should be added to identify the curves. The number of ticks at the x-axis should be increased to make the interpretation easier.

Line 171: „The weight loss with maximum at 619 °C is attributed to thermal decomposition of calcite” – Firstly, the weight loss of each step should be marked. As the waste diatomite contains only 0.2 wt% calcite, I am not convinced that the thermal decomposition of calcite (which causes about 0.09% weight loss) is detectable. What is more, calcite typically starts to decompose at 700°C. According to other researchers, the weight loss at this temperature range is caused mainly by the loss of bound water or the decomposition/burnout of inorganic compounds (see: https://link.springer.com/article/10.1007/s10973-020-10015-3  , and https://link.springer.com/article/10.1007/s10717-014-9641-y ). Later (lines 229-230), „thermal decomposition of organic matter” is marked as the reason for color changes in the fired samples. Why isn’t this process identified in the TG-DTA evaluation?

Figure 7: By what device were the images taken?

There are some editing errors in the text, these are highlighted with yellow in the reviewed manuscript.

Reviewer 5 Report

Dear Authors

The article was written in a transparent manner. However, I have a few technical notes:

  • line 60, 115, 141, 154, 170 (2), 172 (3), 206, 232, 282 - Use subscript instead of a smaller font for chemical formulas,
  • line 71 after than space,
  • line 79 after by space,
  • Figure 1 Please describe the graph in more detail. Mark the temperatures in the diagram
  • line 170 (2) remove "," at the end of the chemical reaction 
  • line 172 (3) remove "." at the end of the chemical reaction

Best Regards

Reviewer 6 Report

Dear Authors,

Thank you very much to give me the possibility to read this interesting manuscript related to the potential of use of waste diatomaceous earth from production of diatomaceous earth for filtration purposes, as an alternative raw material for foam glass production. The manuscript is well structured and clearly describe research questions. It shows good level of coherence between its aim and the results achieved and adequate is the references citated.  To be more in line with the aim of the special issue titled “Advanced Designs of Materials, Machines and Processes in a Circular Economy”, I suggest focusing on circular economy approach (CEA). Lines 38-56, you have made appropriate considerations but, probably, you can better tie them to CEA in the field of material management in technological processes. Moreover, you can improve the conclusion better discussing what could be the role of waste diatomite as alternative raw material for foam glass ceramics production.

Overall, I am convinced that the manuscript is potentially of publishable standard after a minor revision.

Kind Regards

Round 2

Reviewer 2 Report

The manuscript can accept.

Reviewer 3 Report

This manuscript has been revised in detail. It can be accepted for publishing after the further improvement of English language and style.